# Migrating mule deer compensate en route for phenological mismatches

Anna C. Ortega[1,2] ✉, Ellen O. Aikens ®[3], Jerod A. Merkle[4], Kevin L. Monteith[1,5] & Matthew J. Kauffman[6]

Billions of animals migrate to track seasonal pulses in resources. Optimally timing migration is a key strategy, yet the ability of animals to compensate for phenological mismatches en route is largely unknown. Using GPS movement data collected from 72 adult female deer over a 10-year duration, we study a population of mule deer (*Odocoileus hemionus*) in Wyoming that lack reliable cues on their desert winter range, causing them to start migration 70 days ahead to 52 days behind the wave of spring green-up. We show that individual deer arrive at their summer range within an average 6-day window by adjusting movement speed and stopover use. Late migrants move 2.5 times faster and spend 72% less time on stopovers than early migrants, which allows them to catch the green wave. Our findings suggest that ungulates, and potentially other migratory species, possess cognitive abilities to recognize where they are in space and time relative to key resources. Such behavioral capacity may allow migratory taxa to maintain foraging benefits amid rapidly changing phenology.

Each year, animals worldwide migrate across vast landscapes and seascapes to exploit seasonal resources, escape severe weather, breed, avoid predation, or derive other benefits[1]. To maximize energetic gain during migration, many animals synchronize their movements with ephemeral peaks in resource quality or abundance[2–4]. Resource tracking is a key benefit of migration because it promotes nutritional gain, survival, and reproductive success[5,6]. To match their movements with resource phenology, migratory animals often rely on changes in local conditions (i.e., proximate cues)[2,3]. For example, some Neotropical birds use the flowering phenology of honey mesquite (*Prosopis glandulosa*) to initiate their spring migrations with peaks in arthropod abundance[7]. Proximate cues, however, may fail to reflect resource phenology further along a migratory route or distant seasonal range, which can cause animals to become mismatched from peaks in resource phenology during migration[2,3]. When individuals lack the behavioral plasticity to alter their movements en route, migratory taxa can suffer reduced demographic performance and population decline from phenological mismatch[8–10].

Climate change is altering patterns of resource phenology, while the expanding human footprint can decouple animal migration from key resources[11,12]. Together, these rapid environmental changes are expected to cause dramatic, and potentially detrimental, phenological mismatches for migratory taxa[3,5,11]. For example, birds, ungulates, ursids, fish, and cephalopods are altering the timing of migration in response to climate change but at different rates than their primary food[5,11,13]. Adjusting movement en route may be the most rapid and least costly way for many migratory taxa to behaviorally compensate for phenological mismatches, facilitating their adaptation to a rapidly changing world[3,11] (but see[14,15]). Behavioral compensation is likely beneficial for migratory animals that only have a short time to capitalize on fleeting resources, because without compensation, dramatic changes in weather, resource phenology, and reliability of cues would

[1]Wyoming Cooperative Fish and Wildlife Research Unit, Department of Zoology and Physiology, University of Wyoming, Laramie, WY 82071, USA. [2]Program in Ecology, University of Wyoming, Laramie, WY 82071, USA. [3]U.S. Geological Survey, South Dakota Cooperative Fish and Wildlife Research Unit, Department of Natural Resource Management, Brookings, SD 57006, USA. [4]Department of Zoology and Physiology, University of Wyoming, Laramie, WY 82071, USA. [5]Haub School of Environment and Natural Resources, University of Wyoming, Laramie, WY 82072, USA. [6]U.S. Geological Survey, Wyoming Cooperative Fish and Wildlife Research Unit, Department of Zoology and Physiology, University of Wyoming, Laramie, WY 82071, USA. ✉e-mail: aortega1950@gmail.com

carry substantial fitness costs[3,11]. Most previous studies that have explored behavioral compensation during migration, however, focus on the ability of animals to compensate for spatial drift when cross-winds or ocean currents push animals off their migratory trajectory[16]. Aside from a few examples in migratory birds[17,18], the ability of animals to compensate when they drift temporally rather than spatially off course remains undocumented for most migratory taxa[11,19].

Large herbivores, including ungulates, acquire high-quality forage by tracking fleeting waves of emerging plants during migration, which is known as "green wave surfing" (i.e., the Green Wave Hypothesis[20,21]). By tracking the wave of green-up across a landscape, migrating ungulates can continually access new plant growth that is highly nutritious and easy to digest[21]. Previous research indicates that temperate ungulates often initiate spring migration when green-up begins on winter range, allowing them to track peaks in forage quality during spring migration[22-24]. Strong behavioral responses to plant phenology often facilitate synchronous departure from winter range among individuals[22,23,25]. Asynchronous migration may occur, however, when changes in local environmental conditions are subtle and do not correspond to conditions occurring further along the migration trajectory (i.e., the resource gradient is weak, noisy or absent at the beginning of migration)[25]. When conditions change and cues become unreliable, the ability of animals to resynchronize their movements with resource phenology will likely underpin the foraging benefits of spring migration.

To study behavioral compensation for phenological mismatches en route, we took advantage of a unique system where mule deer (*Odocoileus hemionus*) wintering in a desert ecosystem initiated spring migration anywhere from 1–103 days apart. From 2011–2020, we collected GPS movement data from 72 adult female mule deer (>1-yr-old) that spent winter in the Red Desert of south-central Wyoming, USA and migrated long distances (134–293 km) to high-elevation summer ranges (Fig. 1a). Over the 10-year duration of the study, deer often began

migration far ahead or far behind the green wave (Figs. 2 and 3), allowing us to investigate if and how deer resynchronize their movements with peak green-up en route.

## Results

### Triggers of spring migration and cue reliability

Long-distance migrants departed winter range asynchronously with some individuals starting spring migration in mid-February (2 months ahead of peak forage quality) and other individuals starting spring migration in early June (1.5 months behind peak forage quality; Figs. 1b, 2; Supplementary Tables 1, 2). Migration distance and intrinsic factors, including age and pregnancy, did not explain variation in the start of spring migration (although nutritional condition weakly influenced when deer departed their winter range; Supplementary Fig. 1; Supplementary Table 3).

We tested for potential environmental cues that could influence an animal's decision to migrate and found weak responses to plant phenology and temperature (Supplementary Discussion; Supplementary Table 4). To quantify cue reliability across space, we examined the rate of change in green-up along the migratory route. In 5 of the 8 tracking years where analysis was possible, there was either a negative rate of change in green-up or no correlation between the date of peak green-up and distance over the first 32 km of the migration ($p = 0.08–0.92$), meaning that the green wave was nonexistent or propagated backwards over this segment of the route (Fig. 4a; Supplementary Table 5). Thus, on the desert winter range and during early portions of the 240-km migration, spring green-up does not move as a wave, making it difficult to track even if local environmental cues were used to initiate migration. Although the underlying mechanisms that influence the start of spring migration are unclear, mule deer departed winter ranges at markedly different times relative to peak green-up. The unique variation among individuals in the onset of migration creates the opportunity to evaluate whether deer can perceive their

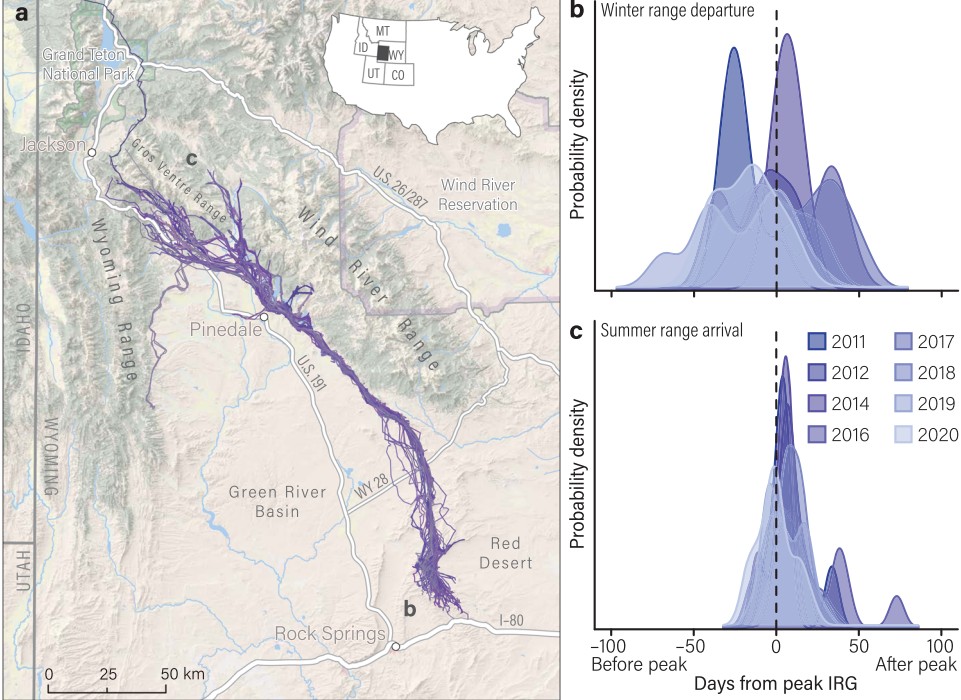

**Fig. 1 | Probability density of mean days from peak Instantaneous Rate of Green-up (IRG) at the start and end of spring migration. a** Each spring, mule deer leave their desert winter range to make a 240-km, one-way, migration in western Wyoming. **b** The start date of spring migration was standardized to the mean date of peak IRG on winter ranges for each year of the study (2011–2012, 2014,

2016–2020). The vertical dashed line represents zero days from peak IRG. **c** The end date of spring migration was standardized to the mean date of peak IRG on summer ranges for each year. Mule deer started spring migration asynchronously (70 days ahead to 52 days behind peak IRG) but, on average, arrived on summer range within a narrow window of time (6 days).

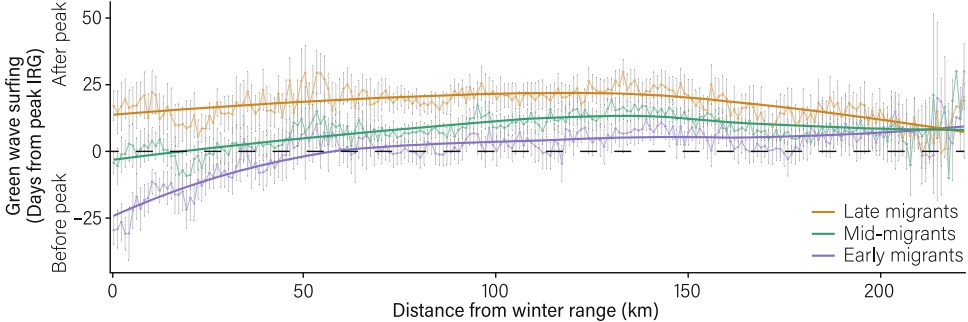

**Fig. 2 | Green wave surfing en route for early, mid and late migrants.** Green wave surfing en route was measured as mean days from peak Instantaneous Rate of Green-up (IRG) ± 95% confidence intervals as a function of distance from winter range (solid lines represent loess regressions). Despite being strongly mismatched ahead or behind the green wave when they began their spring migration, early migrants ($n$ = 47 animal-years; purple), mid-migrants ($n$ = 58 animal-years; green), and late migrants ($n$ = 47 animal-years; orange) ended spring migration largely synchronized and closer to peak IRG. Early and late migrants seemingly compensated for being mismatched with the green wave during migration. The horizontal dashed line represents perfect surfing (0 days from peak IRG).

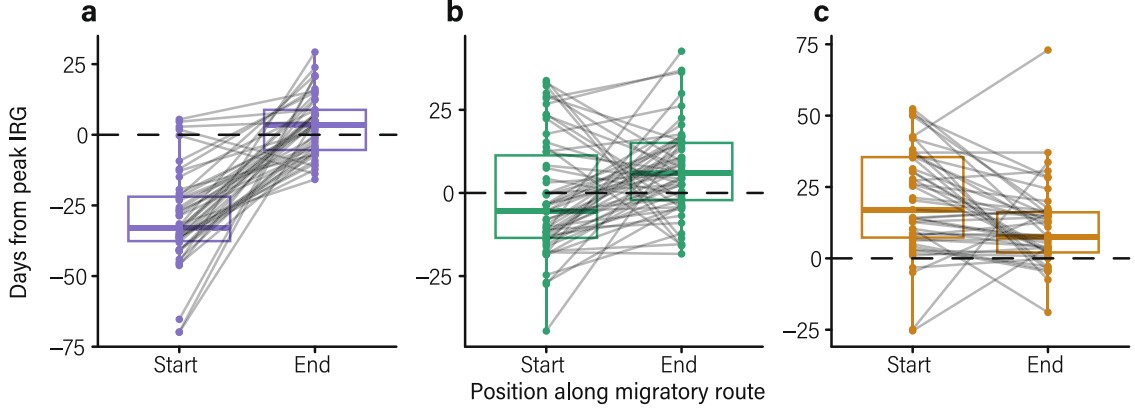

**Fig. 3 | Comparisons in the location of mule deer on the green wave at the start and end of spring migration. a** Despite starting spring migration 30 ± 5 days ($\bar{x}$ ± 95% CI) ahead peak Instantaneous Rate of Green-up (IRG; dashed horizontal line), early migrants ($n$ = 47 animal-years; purple) ended their spring migration 4 ± 3 days behind peak IRG. **b** Mid-migrants ($n$ = 58 animal-years; green) were 2 ± 5 days ahead peak IRG at the start of spring migration but 7 ± 3 days behind peak IRG at the end of spring migration. **c** Although late migrants ($n$ = 47 animal-years; orange) started spring migration 20 ± 5 days behind peak IRG, they ended spring migration 11 ± 4 days behind peak IRG. The boxplots represent the median value of days from peak IRG (horizontal bar). Whiskers extend to the minima (25th percentile – 1.5 * interquartile range) and maxima (75th percentile + 1.5 * interquartile range).

own location in space and time relative to the green wave (e.g., ahead or behind peak green-up) and behaviorally adjust their movements en route to compensate for initial phenological mismatches.

**Behavioral compensation en route**

Using the distribution of the start date of spring migration, we classified deer as early migrants (≤25% quartile; $n$ = 47 animal-years), mid-migrants (>25% and <75% quartiles; $n$ = 58 animal-years), and late migrants (≥75% quartile; $n$ = 47 animal-years; Supplementary Table 6). Early migrants traveled 186 ± 8 km ($\bar{x}$ ± 95% CI) over a period of 72 ± 6 days. Mid-migrants traveled 189 ± 5 km over a period of 48 ± 4 days. Late migrants traveled 188 ± 6 km over a period of 31 ± 5 days. Of the $n$ = 48 deer that had more than two years of GPS data, 60% of deer ($n$ = 29) were not consistent across years in whether they were early, mid, or late migrants.

We evaluated whether mule deer compensated for phenological mismatches with the green wave by comparing the synchronicity of an individual's movements to the green wave at both the start and end of migration, and during the entire spring migration. We calculated the difference in days between the date of each GPS location and the date of peak Instantaneous Rate of Green-up (IRG) at each location along the route (hereafter referred to as Days-From-Peak[21]). According to the

Green Wave Hypothesis, ungulates are assumed to obtain maximum foraging benefits when they occupy habitats on the day of peak IRG (Days-From-Peak = 0)[26].

Early migrants started spring migration 30 ± 5 days ($\bar{x}$ ± 95% CI) ahead of the green wave but began to match their movements with peak forage quality approximately 57 km from their winter range (Fig. 2). By the time they arrived at their summer range, early migrants were only 4 ± 3 days behind the green wave, 26 days closer than when they started spring migration (paired-sample $t$ test, $t_{46}$ = −10.65, $p$ = 5.37 × 10$^{-14}$). Late migrants, on the other hand, started spring migration 20 ± 5 days behind the green wave and began to catch up with peak forage quality approximately 121 km into the migration (Fig. 2). Late migrants arrived on their summer range 11 ± 4 days behind the green wave, which was 9 days closer than when they started spring migration (Wilcoxon signed-rank test, $Z$ = −3.50, $p$ = 0.0005). Mid-migrants started spring migration 2 ± 5 days ahead of the green wave but ended spring migration 7 ± 3 days behind the green wave (paired-sample $t$ test, $t_{57}$ = −2.91, $p$ = 0.005; Fig. 3). On average, early and mid-migrants were 7 days closer to peak green-up than late migrants during spring migration (ANOVA, $F_{2,149}$ = 15.59, $p$ = 7.14 × 10$^{-7}$). Although early migrants started spring migration 24 days before mid-migrants and 45 days before late migrants, all mule deer completed their spring

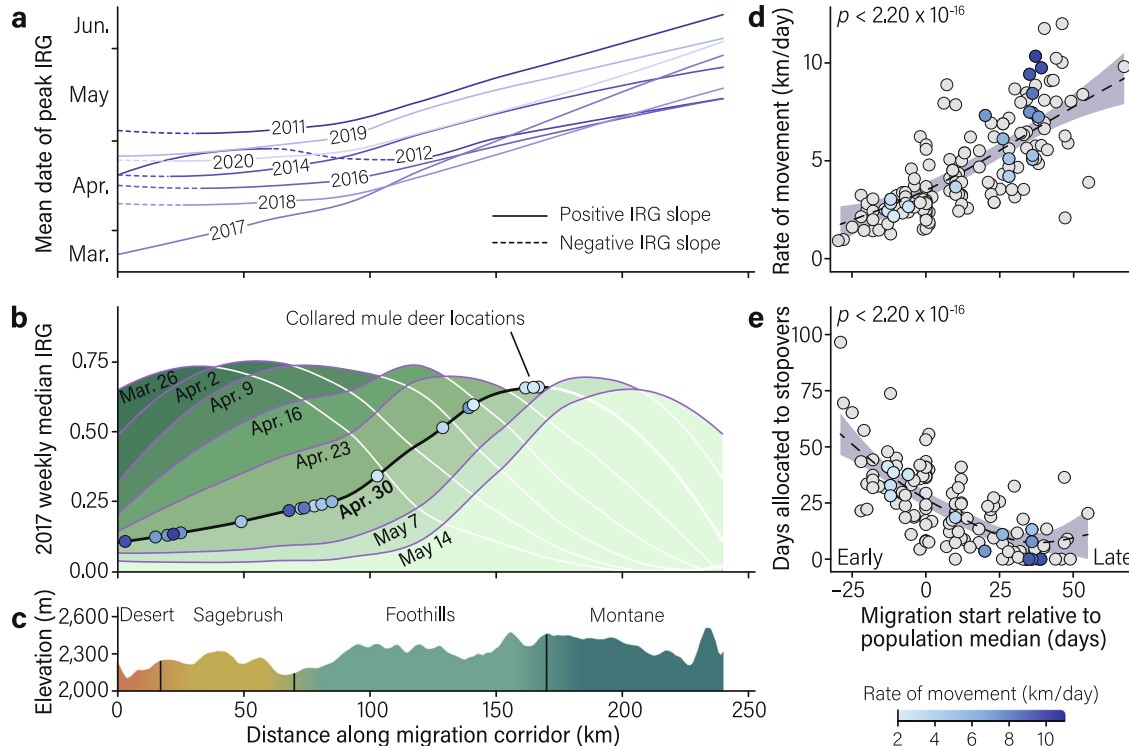

**Fig. 4 | Movement rate and stopover use by migrating mule deer relative to the propagation of the green wave across western Wyoming. a** Plotting mean date of peak Instantaneous Rate of Green-up (IRG) along the 240-km migration corridor indicates that the wave of green-up did not propagate consecutively across the landscape for the first 32 km of migration (dotted segments indicate a negative slope of green-up) but propagated as a wave for the remainder of the migration corridor (solid segments indicate a positive slope of green-up). **b** The spring of 2017 illustrates the movements of full compensators (blue dots) relative to the weekly propagation of the green wave. Mule deer that were behind the green wave tended to move more quickly, while those ahead of the wave moved more slowly. **c** Mule deer migrated 134–293 km over 50 ± 4 ($\bar{x}$ ± 95% CI) days from a desert sagebrush shrubland to a montane ecosystem. **d** The start date of spring migration (i.e., whether deer were early or late) positively influenced the rate of movement by mule deer during spring migration (predicted coefficients ± 95% CI; GAMM, $R^2 = 0.56$, $p < 2.20 \times 10^{-16}$). Colored points correspond with the same individuals in b and indicate the average rate of movement over the entire migration for those individuals. **e** The start date of spring migration negatively influenced the number of days mule deer allocated to high-use stopovers (predicted coefficients ± 95% CI; GAMM, $R^2 = 0.53$, $p < 2.20 \times 10^{-16}$). The start date of spring migration for each animal-year was standardized relative to the median start date of spring migration for each year of the study.

migration, on average, within a 6-day window (Supplementary Fig. 2; Supplementary Table 7). Whether the timing of green-up was early or late, mule deer arrived at summer range closer to peak green-up than when they started spring migration (Supplementary Table 8). Thus, mule deer are behaviorally flexible in the pace of their migration and appear to have a strong ability to compensate en route for phenological mismatches with peak forage quality, allowing them to resynchronize their movements with the green wave.

In the xeric habitats of the Red Desert, the green wave propagates gradually, but green-up becomes more rapid and fleeting as it progresses towards the mountainous summer range (Fig. 4a, c). Because of this, the foraging penalty for being mismatched (which is highest when green-up is most rapid[21]) was 11% higher during the last quarter of migration than the first quarter of migration ($p < 2.20 \times 10^{-16}$; Supplementary Fig. 3). The ability of mule deer to compensate for phenological mismatches en route thus reduces the foraging penalty they would have experienced if mismatched during later parts of their migration where green-up is more wave-like.

We investigated two complimentary behavioral mechanisms of compensation: movement rate and time spent on stopovers. How fast deer moved along their migratory route was strongly linked to whether they were ahead or behind the green wave when they started migration. Despite being overall worse surfers than early and mid-migrants, late migrants made the best of their strategy by accelerating their movement. In contrast to early migrants that moved slowly (2.9 ± 0.3 km/day) and spent extended time on stopovers (36 ± 5 days),

late migrants moved 2.5 times faster (7.1 ± 0.7 km/day) and spent 72% less time on stopovers (10 ± 5 days; ANOVA$_{rate}$, $F_{2,149} = 66.40$, $p = 2.40 \times 10^{-21}$; ANOVA$_{stopover}$, $F_{2,124} = 33.55$, $p = 2.26 \times 10^{-12}$; Fig. 4d, e), allowing them to catch up with the green wave. Migration distance, age, pregnancy, and nutritional condition did not influence how fast or slow mule deer migrated (Supplementary Tables 9, 10), suggesting that movement rate and stopover use are largely linked to an animal's position along the green wave. Mule deer seemingly gathered information along their migratory route to recognize their own temporal deviation from peak green-up and made movement decisions that allowed them to catch up to the green wave—or allowed the wave to catch up with them.

### Variation in behavioral compensation

Mule deer varied in their ability to compensate for mismatches with the green wave during migration (Supplementary Table 11). Ninety-three percent of mule deer that started spring migration ahead of the green wave fully (70%) or partially (23%) compensated for phenological mismatches by decelerating their movement, whereas 90% of mule deer that started spring migration behind the green wave fully (54%) or partially (36%) compensated for phenological mismatches by accelerating their movement (Fig. 4b; Supplementary Fig. 4). Moreover, mule deer that were further from peak green-up at the onset of spring migration were more likely to become full compensators. For every 1-day increase in mismatch with the green wave, mule deer were 1.21 times more likely to fully compensate ($\beta = 0.19$, 95% CI = 0.15–0.24,

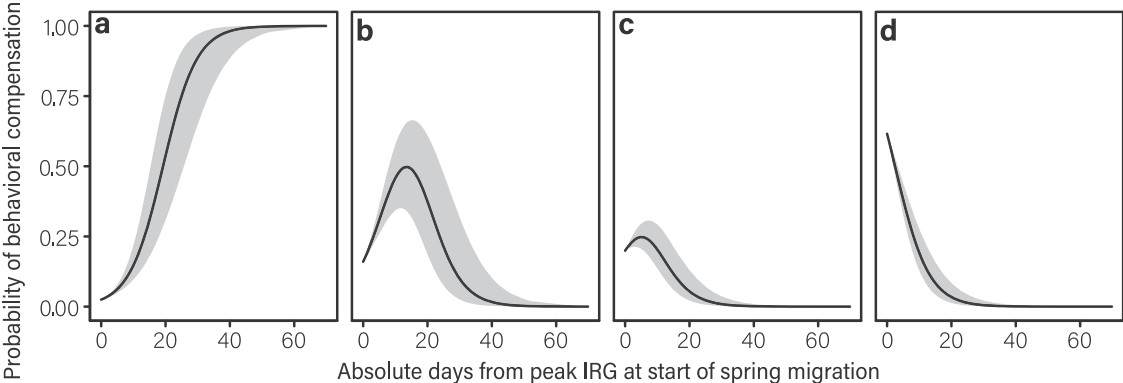

**Fig. 5 | The probability of mule deer to behaviorally compensate based on their degree of mismatch with the green wave at the start of spring migration.** Mule deer were classified as (**a**) full compensators, (**b**) partial compensators, (**c**) perfect surfers, and (**d**) non-compensators. Based on an ordinal logistic regression model, mule deer were 1.21 times more likely to become full compensators for every 1-day increase in mismatch with peak Instantaneous Rate of Green-up at the start of spring migration (β = 0.19, 95% CI = 0.15–0.24, p = 6.98 ×10⁻¹⁵). Beyond 40 days mismatched, nearly 100% of mule deer behaviorally compensated to realign their movements with the green wave. Grey bands represent the 95% confidence intervals of the predicted probabilities.

$p = 6.98 \times 10^{-15}$; Fig. 5). Nonetheless, 15% of mule deer were non-compensators and became further mismatched from peak forage quality during migration. The overall ability of compensators to accelerate or decelerate their movement underscores the importance of keeping pace with the green wave[27].

Migratory performance, including the ability to track resources, can improve through individual learning or social learning and cultural transmission[28,29]. We incorporated age into our analyses to evaluate if individual learning and experience influenced the ability of mule deer to compensate for phenological mismatches with the green wave. There was no effect of age (range: 1–13 years old) on the probability of individuals being full compensators (β = −0.04, 95% CI = −0.16 to 0.08, p = 0.52).

## Discussion

Our understanding of the ability migratory animals have to compensate for being mistimed with resource phenology is still in its infancy. The limited previous research on temporal compensation in resource tracking is taxonomically biased toward avian migrations[17,18,30]. For example, some Nearctic-Neotropical birds and barnacle geese (*Branta leucopsis*) adjust their movement rate during migration, allowing them to arrive at their breeding range when forage quality is at its peak[17,18]. Few studies, however, have documented an animal's ability to compensate continuously along the length of their migration[3,11,19]. By combining fine-scale movement data with detailed estimates of the green wave in space and time, we found that migrating ungulates are behaviorally flexible and have the capacity to readily adjust their speed en route. This study broadens our understanding of the behavioral plasticity that allows diverse migratory taxa to track the dynamic nature of resources across space and time. The ability to behaviorally compensate en route should therefore be extended beyond migratory animals that experience spatial drift[16] and include migratory animals that experience temporal mismatches[11] from peaks in resource phenology. The adaptive capacity to compensate for phenological mismatches may allow some migratory animals to maintain foraging benefits amid rapid changes in environmental conditions.

Behavioral compensation may be particularly important for animals that adopt an income migration strategy in which they forage extensively during migration (as opposed to capital migrants that rely on stored energy reserves during migration[31]). Mule deer rely on both capital and income resources, using fat reserves from previous seasons and forage during migration, to finance reproduction on summer range[32]. By being behaviorally plastic during migration, mule deer can time their arrival on summer range with the green wave, which may

help ensure that the energetically taxing period of parturition coincides with peaks in forage quality[32]. Indeed, animals with an income-based migration strategy are expected to be behaviorally plastic in their migration schedule, because they can continuously acquire information on their food resources throughout the length of their migration[31].

A key finding of this study is that mule deer have the cognitive capacity to recognize when they are temporally mismatched from the green wave. The precise mechanism of how deer identified their location on the green wave remains unclear, although passage rates of forage through the digestive tract, neurological responses to gut capacity, and changes in the rate of forage intake could provide deer with the necessary cognitive information[33-35]. For example, mature, fibrous forbs remain 2.4 times longer in a mule deer's digestive tract than immature forbs[33]. As foraging ruminants reach gut capacity, the secretion of leptin from white adipose tissue produces a feeling of satiety, reducing the voluntary intake of forage[34,35]. Slower or faster rates of passage and disparate intake of forage, along with visual or olfactory perception, and spatial memory of resource phenology[36], could easily provide a reliable cue of a migrating animal's position relative to the green wave. Regardless of the sensory mechanisms involved, this work highlights the temporal cognitive ability of migratory animals, which is vastly underrepresented in research on animal migration compared with spatial cognitive abilities[16,29,37]. Furthermore, these results call into question an often overly simplistic view of animal cognition that underlies theories of trophic mismatch.

Mule deer are gregarious animals and typically migrate in groups of variable sizes and age classes[38,39], which could allow group decision-making to determine when to start migrating and how to keep pace with the green wave during migration. The ability of young individuals to compensate as well as older individuals suggests that either social learning or collective navigation likely influence the ability of mule deer to compensate for phenological mismatches during migration. Social learning from conspecifics may improve the ability of animals to efficiently navigate and track resources en route. For example, naïve whooping cranes (*Grus americana*) learned from older, experienced individuals to pace their seasonal migrations with the green-up of plants in spring and onset of snow in autumn[29,40]. If social learning improves an animal's ability to track key resources and enhance fitness, then sociality may allow some migratory animals to more easily cope with rapid environmental change.

Our research focused on an ungulate population that migrates across a relatively intact landscape with navigable fences and only two rural highways intersecting the 240-km migration corridor[39] (Fig. 1a).

The existence of few impediments or other behavioral disruptions likely facilitated the ability of mule deer to compensate for phenological mismatches with the green wave during migration[41]. Barriers that impede animal movement, such as high-traffic interstates, impermeable fences, or housing developments, interfere with an animal's decision-making during migration and dampen their behavioral plasticity[42–44]. For example, migrating animals often accelerate through development to minimize disturbance and may stall for extended periods of time before entering development, which complicates their ability to keep pace with resource phenology[12,23,45]. When considering the ability of animals to behaviorally compensate during migration, ecologists must be cognizant of barriers that constrain the free movement and resource tracking of animals. Conserving intact landscapes is thus necessary to ensure that animals have the greatest opportunity to adapt their movement strategies to changing patterns of the fleeting resources they seek.

## Methods

### Animal capture and handling

From 2014–2020, we captured $n = 220$ adult female mule deer (>1-yr-old) in the Red Desert near Rock Springs, Wyoming, USA (41° 35′N, 109° 12′W) as part of a long-term study. We recaptured deer each March and December for a total of $n = 528$ animal-years of data (Supplementary Methods). All deer were captured via helicopter net-gunning[46,47]. Mule deer in this portion of the Sublette Herd migrate a variety of distances to their summer ranges in northwestern Wyoming[39,48]. Herein, we focused on $n = 72$ long-distance migrants ($n = 152$ animal-years) that migrated 134–293 km and spent the summer north of Pinedale, Wyoming (42° 51′N, 109° 51′W). During captures, we used an electronic platform scale (±0.1 kg) to measure body mass (kg) and a portable ultrasound (Ibex, E.I. Medical Imaging, Loveland, CO) to measure maximum rump fat (mm). Following previously applied methods[49], we used body mass, maximum rump fat, and a body-condition score to estimate percent-scaled ingesta-free body fat (IFBFat)[49,50]. For captures in March, we used an ultrasound to determine pregnancy, including fetal rate (number of fetuses per deer) and fetal development via measures of the fetal eye diameter (mm). To estimate the age of each deer, we extracted the lower right incisiform canine and used the cementum annuli aging technique[51–53], which was conducted by Matson's Laboratory in Manhattan, Montana, USA. From 2014–2020, we outfitted all deer with store-on-board or iridium GPS collars that collected locations every 1–2 hrs (Advanced Telemetry Systems, Isanti, MN, USA; Lotek Wireless, Newmarket, ON, CAN; Telonics, Mesa, AZ, USA). We also included GPS collar data from a previous study on the Sublette Mule Deer Herd (2011–2013)[48] to analyze movement for $n = 27$ additional deer ($n = 66$ animal-years), which were outfitted with store-on-board GPS collars that collected locations every 3 h (Telonics, Mesa, AZ, USA). All animal capture and handling protocols were approved by the Wyoming Game and Fish Department (Chapter 33-937) and an Institutional Animal Care and Use Committee at the University of Wyoming (20131111KM00040, 20151204KM00135, 20170215KM00260, 20200302MK00411). Data associated with the capture and handling of mule deer were organized and managed within Microsoft Access (Microsoft Office Version Professional Plus 2016).

### Delineation of migratory routes and seasonal ranges

We used Net Square Displacement (NSD)[54] to determine the timing of spring and autumn migration, delineate migratory routes, and determine the net displacement (km) between the start of spring migration and each GPS location along the migratory route. We determined winter range use for each deer by extracting GPS locations between the end of autumn migration and the start of spring migration (or between the time of capture and start of spring migration if the end of autumn migration was unknown). We determined summer range use for

each deer by extracting GPS locations between the end of spring migration and the start of autumn migration (or between the end of spring migration and time of collar failure or mortality on summer range). We used a 95% Kernel Utilization Distribution (KUD)[55] to delineate the winter range (41.63 ± 7.26 km² [$\bar{x}$ ± 95% CI]) and summer range (7.26 ± 1.61 km²) of each animal-year. We removed 0.10%, 0.09%, and 0.24% of all GPS locations during migration, on winter range, and on summer range, respectively, because the movement rate between consecutive locations was greater than 10.8 km/h and indicated an inaccurate GPS fix.

### Factors influencing the start of spring migration

We sought to identify intrinsic and extrinsic factors that could explain variation in the start of spring migration among mule deer. We conducted a generalized linear mixed model (GLMM) to understand the effect of intrinsic variables, including age, nutritional condition (% scaled IFBFat in March), and pregnancy (i.e., fetal rate [number of fetuses per deer]; fetal eye diameter [mm]), in addition to migration distance (km), on the start of spring migration. We standardized the start date of spring migration for each animal-year to the median start date of spring migration for each year of the study. For the GLMM and all other mixed models in our analyses, we used year and animal ID as random intercepts to account for climatic variability across years and repeated measures of the same individual in different years. We used the drop1 function in the R stats package to identify the significance of each fixed effect and select the best-fitting model. We performed the GLMM with the lme4 package[56] within R version 4.0.5[57].

### Cue-responses on winter range

We evaluated how changes in environmental conditions on winter range (i.e., plant phenology, temperature, snow depth, photoperiod) initiated spring migration for mule deer. We used the Normalized Difference Vegetation Index (NDVI) from a time series of MODIS satellite images (MOD09Q1 satellite array; 250-m² spatial resolution, 8-day temporal resolution) to quantify changes in plant phenology on winter range. NDVI values were fit to a double logistic curve and scaled between 0 and 1 with values of 1 indicating maximum plant biomass for a given pixel on the landscape[58]. We used NDVI as a metric of plant phenology rather than Instantaneous Rate of Green-up (IRG) because IRG is symmetrical around its peak[58] and can be difficult to interpret linearly. We used spatial climate data from the Parameter-elevation Regressions on Independent Slopes Model (PRISM; Climate Group at Oregon State University, USA) to estimate mean daily temperature (°C; 4-km² spatial resolution, 1-day temporal resolution)[59]. We used the Snow Data Assimilation System (SNODAS; National Snow and Ice Data Center) to estimate snow depth (m) across the study area (1-km² spatial resolution, 1-day temporal resolution)[60]. We determined photoperiod (time of sunrise−time of sunset) for each GPS location with the R suncalc package[61]. We extracted NDVI, temperature, snow depth, and photoperiod for each GPS location prior to spring migration and calculated the daily mean of NDVI, temperature, snow depth, and photoperiod for each animal-year to reduce pseudoreplication and account for irregular GPS fixes. We also calculated the daily changes in NDVI, temperature, and snow depth for each animal year.

We used a mixed effects Cox Proportional Hazards (CPH) model[62,63] to evaluate the effect of changes in plant phenology, temperature, snow depth, and photoperiod on the likelihood of deer initiating spring migration. We subset GPS locations from 20 March to the start of each individual's spring migration to standardize time series among all animal-years, including $n = 43$ deer that we captured in mid-March. We used year and animal ID as a random intercept and each environmental variable (i.e., plant phenology, temperature, snow depth, photoperiod) as a random slope, which allowed us to obtain random effect coefficients (i.e., individual cue-response) of environmental variables on the daily instantaneous rate that each individual

would start spring migration. We validated the key assumption of the CPH model that the baseline hazard remained proportional for each covariate (Supplementary Methods)[63–65] and conducted an Additive Cox Proportional Model to verify that all predictor variables exhibited a linear relationship with the hazards ratio (elective degrees of freedom [e.d.f] = 1.0). We then selected the best-fitting model for fixed effects using backward stepwise selection, and adopted a final model containing the fixed and random effects with the survival package[66] within R version 4.0.5[57].

## Cue reliability: propagation of the green wave across space
We evaluated the annual propagation of the green wave across the study area by calculating the mean date of peak IRG for each kilometer along the 240-km migration corridor. First, we calculated the 99% utilization distribution (UD) from a Brownian bridge movement model (BBMM[67,68]) to delineate the migration corridor for each animal-year and then merged all migration corridors into one polygon to delineate the population-level migration corridor from 2011–2020. For each year of the study, we extracted date of peak IRG to the population-level migration corridor. We calculated the Euclidean distance between each pixel on the landscape and the southernmost location of the migration corridor (41° 41′N, 108° 50′W) and only included pixels that had a Euclidean distance of less than or equal to 240 kilometers to limit our analyses within the designated 240-km migration corridor[39]. We calculated rate of change in mean date of peak IRG and categorized rate of change as either negative or positive. Negative values indicated a non-consecutive green wave that propagated backwards towards winter range, whereas positive values indicated a consecutive wave that propagated northward towards summer range. We then conducted a series of linear regressions for each year and direction of the green wave (negative versus positive) to further understand whether the order of green-up was consecutive. Each linear regression included distance along the migration corridor as the predictor variable and mean date of peak IRG as the response variable. A lack of correlation between distance and date of green-up indicated that green-up did not propagate like a wave across the landscape.

Based on the linear regression between mean date of peak green-up and distance along the migration corridor, the spring of 2017 exhibited the strongest and most consecutive propagation of green-up across the landscape (Supplementary Table 5). Thus, we used IRG from 2017 to visualize the weekly propagation of the green wave. We cropped MODIS satellite images containing daily IRG values in 2017 to the population-level migration corridor. From March 26, 2017 through May 20, 2017, we calculated the median IRG for each kilometer and week. We then used a loess regression to plot the median IRG for each kilometer and week along the migration corridor.

## Green wave surfing
We evaluated the ability of mule deer to track green-up of plants during spring migration by analyzing the synchronicity between movement and peak IRG. We determined IRG by extracting the first derivative of double-logistic curves that were fit to the annual time series of NDVI[21]. Days from peak IRG (hereafter referred to as Days-From-Peak) were calculated as the difference in days between the date of every GPS location for a deer and the date of peak IRG at the same GPS location[6,21]. A theoretically perfect surfer occupies a location on the same day that peak IRG occurs (Days-From-Peak = 0)[21]. We calculated mean Days-From-Peak for each day and kilometer of an individual's migration to reduce pseudoreplication and account for irregular GPS fixes[6,21]. We quantified an individual's location on the green wave at the start and end of spring migration by calculating the difference in days between the mean date of peak IRG on each seasonal range and the date an individual departed their winter range or arrived at their summer range.

## Annual variation in phenological mismatches with the green wave
We evaluated whether mule deer at the population level experienced phenological mismatches with peak IRG at the start and end of spring migration. First, we standardized the start of spring migration for each deer by subtracting the start date of spring migration from the mean date of peak IRG on each winter range. Second, we standardized the end of spring migration for each deer by subtracting the end date of spring migration from the mean date of peak IRG on each summer range. We performed a series of two-sided z-tests to determine whether deer started and ended spring migration ahead or behind peak IRG.

## Behavioral compensation
We determined whether early, mid, and late migrants compensated for phenological mismatches with the green wave during migration by conducting a series of paired $t$ tests that compared an individual's location on the green wave at the start and end of spring migration. We also conducted a one-way multivariate analysis of variance (MANOVA) to compare start and end dates of spring migration and Days-From-Peak at the start and end of spring migration among early, mid, and late migrants. The MANOVA included departure from winter range (i.e., early, mid, late) as the independent variable and the following dependent variables: (1) start date of spring migration, (2) end date of spring migration, (3) Days-From-Peak at the start of spring migration, and (4) Days-From-Peak at the end of spring migration. We then conducted four separate Tukey's honestly significant different (HSD) tests as *post hoc* analyses to further identify differences in timing and Days-From-Peak at the start and end of spring migration among early, mid, and late migrants.

We identified mule deer as full compensators, partial compensators, perfect surfers, and non-compensators based on how mismatched they were from peak IRG at the start and end of spring migration (Supplementary Table 11). Full compensators were deer that started spring migration more than 7 days ahead or behind peak IRG but became closer to peak IRG at the end of their spring migration. For full compensators, absolute Days-From-Peak at the end of spring migration was less than absolute Days-From-Peak at the start of spring migration. Partial compensators were deer that started spring migration more than 7 days ahead or behind peak IRG but neither became further from nor closer to peak IRG. Partial compensators did not deviate by more than 7 Days-From-Peak at the end of spring migration compared with the start of spring migration. Perfect surfers were deer that started spring migration within 7 days of peak IRG and remained within 7 days of peak IRG at the end of their spring migration. Non-compensators were deer that drifted further from peak IRG at the end of spring migration regardless of when they departed winter range relative to peak IRG. For non-compensators, absolute Days-From-Peak at the end of spring migration was more than absolute Days-From-Peak at the start of spring migration. We performed two ordinal logistic regression models to evaluate whether absolute Days-From-Peak at the start of spring migration and age influenced the probability of a deer becoming a full compensator versus a partial compensator, perfect surfer, or non-compensator. Our dataset included age for $n = 51$ unique mule deer ($n = 115$ animal-years).

Finally, to understand if deer compensated for phenological mismatches irrespective of whether green-up was early or late, we conducted two simple linear regressions with annual variation in date of peak IRG on winter range as the predictor variable (i.e., annual deviation from long-term average) and absolute Days-From-Peak at the start or end of migration as the response variable.

## Foraging penalties relative to distance and degree of mismatch
We quantified foraging penalties for deer based on how mismatched they were from peak IRG and their location relative to summer range

where green-up was more rapid and fleeting. For each pixel within the migration corridor, we calculated the loss in IRG as the average difference between IRG on the date of peak IRG and IRG from 1 to 7 days before and after peak IRG. We performed a generalized additive mixed model (GAMM) to evaluate the effect of distance along the migration corridor (km) and Days-From-Peak on percent loss in IRG. We divided distance by a factor of 100 to ensure that all covariates were on similar scales. We used an interaction term between distance and Days-From-Peak within the GAMM to understand whether the effect of Days-From-Peak on foraging penalties depended on distance along the migration corridor. All covariates exhibited a strong non-linear relationship with loss in IRG (e.d.f > 3.0). We used a cubic spline to smooth fixed effects and applied shrinkage to smoothed fixed effects, which enabled non-linear fixed effects to be shrunk to zero if they did not significantly contribute to the top model. We used year as a random intercept to reduce pseudoreplication and to account for repeated measures of IRG along the migration corridor. We performed the GAMM with the mgcv package[69] within R version 4.0.5[57].

### Movement rate and stopover use

We determined the movement rate for each deer by dividing the total length of migration (Euclidean distance between start and end locations of each migration) by the duration of migration (days). We used a BBMM with a 150-m resolution and 10% UD to delineate high-use stopovers[70]. We determined the total number of days each deer spent on high-use stopovers (≥3 days) by calculating the quotient between the sum of GPS fixes on each stopover and the GPS fix rate.

We removed $n = 25$ animal-years with BBMM motion variances greater than 8,000. Our final analyses on stopover use included $n = 65$ unique mule deer ($n = 127$ animal-years). Of the $n = 127$ animal-years in the stopover analyses, $n = 112$ animal-years spent some portion of their migration on high-use stopovers (≥3 days), whereas $n = 15$ animal-years spent zero days on high-use stopovers.

We conducted two separate GAMMs to understand the effect of movement rate and stopover use on an individual's ability to compensate for starting spring migration early or late. For each GAMM, we used start of spring migration as the fixed effect and movement parameter (i.e., movement rate or stopover use) as the response variable. We standardized the start date of spring migration for each animal-year to the median start date of spring migration for each year of the study. We used a cubic spline to smooth start of spring migration, which exhibited a weak non-linear relationship with movement rate (e.d.f = 1.96) and a strong non-linear relationship with stopover use (e.d.f = 3.00). We did not conduct model selection beyond the base model because each GAMM contained only one fixed effect. We performed each GAMM with the mgcv package[69] within R version 4.0.5[57]. We then performed a series of one-way ANOVAs and Tukey's HSDs to determine differences in movement rate and stopover use among early, mid, and late migrants and among full compensators, partial compensators, perfect surfers, and non-compensators.

Because reproductive status may influence an animal's movement behavior, including rate of movement during migration[71], we evaluated if other factors besides phenological mismatch influenced movement rate and stopover use. We conducted two separate GLMMs with migration distance (km), age, nutritional condition (% scaled IFBFat in March), and pregnancy (i.e., fetal rate [number of fetuses per deer]; fetal eye diameter [mm]) as the predictor variables, movement rate or stopover use as the response variable, and animal ID as a random intercept.

### Reporting summary

Further information on research design is available in the Nature Portfolio Reporting Summary linked to this article.

## Data availability

Source Data are provided as a Source Data File with this manuscript. Data underlying this research are available in Dryad[72]. Source data are provided with this paper.

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

## Acknowledgements

Funding was provided by the Bureau of Land Management, Hunter Legacy 100 Fund, Knobloch Family Foundation, Muley Fanatic Foundation, National Science Foundation (A.C.O), Pew Charitable Trusts, Safari Club International Foundation, Sitka Ecosystem Grant, Teton Conservation District, The Nature Conservancy, Wyoming Game and Fish Department, Wyoming Governor's Big Game License Coalition, and the US Geological Survey. Hall Sawyer contributed data from 2011–2013 and provided input on the research and manuscript. Ian Freeman and Alethea Steingisser at the University of Oregon InfoGraphics Lab led graphic design of figures. Wildlife biologists at the Wyoming Game and Fish Department (WGFD) and Bureau of Land Management (BLM), including Patrick Burke (WGFD), Dean Clause (WGDF), Patrick Lionberger (BLM), Brandon Scurlock (WGFD), Miguel Valdez (BLM), and Sean Yancey (WGFD) assisted with field logistics and data collection. Any use of trade, firm, or product names is for descriptive purposes only and does not imply endorsement by the U.S. Government.

## Author contributions

A.C.O. and M.J.K. conceived of the study. A.C.O., K.L.M., and M.J.K. collected data from 2014–2020. A.C.O., E.O.A., J.A.M., and M.J.K. developed the methodology. A.C.O. and M.J.K. wrote the manuscript. A.C.O., E.O.A., J.A.M., K.L.M., and M.J.K. contributed revisions to the manuscript.

## Competing interests

The authors declare no competing interests.
