## [Peer Review File · Nature Communications]

Migrating mule deer compensate en route for phenological mismatchesEditorial Note: This manuscript has been previously reviewed at another journal that is not operating a transparent peer review scheme. This document only contains reviewer comments and rebuttal letters for versions considered at *Nature Communications*.

REVIEWERS' COMMENTS

Reviewer #1 (Remarks to the Author):

I have reassessed the manuscript by Ortega et al. as now submitted to Nature Communications. I find the explanations of the authors to my previous queries to be very satisfactory, and their resulting revisions to have resulted in an improved manuscript. I have only two remaining comments. I consider that these comments are rather minor, and that otherwise this manuscript would now be suitable for publication in Nature Communications. I want to congratulate the authors on this interesting and nice manuscript, which is an excellent example on the flexibility of migration strategies.

As the authors rightfully point out, the migration system of birds may be very different from that of ungulates, including differences in how resources are taken up and used during migration. This shows that while similar patterns as found by the authors have been shown in avian species, the findings of the authors are very much novel given that this has not before been shown in ungulates. I think it would be interesting to spend some more discussion on differences in migration systems and strategies. A recent paper by Evans and Bearhop (2022) in the Journal of Animal Ecology pointed out consistent differences between migration strategies with respect to resource use, namely the gradient between 'income migrants' which forage while they travel and 'capital migrants' which store energy reserves prior to departure. Some of the avian migrants which have been shown to be able to compensate for temporal drift belong in the latter category, while mule deer very much belong in the former category. I think it is interesting to discuss the results and potential differences in the ability to compensate for temporal drift between these migration strategies. Maybe this could be discussed in one or several sentences in paragraph 1. 217 - 232.

In my previous review of this manuscript I considered that green-wave surfing en route was not well tested in this manuscript. I thank the authors for their clear explanation that in fact they did test this. On this topic, I have a question or suggestion for figure 2, in which green-wave surfing during migration is shown by displaying Loess regression curves of late, mid- and early migrants. I find that while the Loess curves might work well for visualisation purposes, they do not give a good idea on how the actual data looks like (which might be much more messy indeed), nor are they used to test green-wave surfing as far as I understand from the manuscript. I would suggest to think of alternative ways for displaying the data. Maybe the lines of individual mule deer tracks could be added to the plot.

**Reviewer #2 was unable to check the responses to their comments so reviewer #1 was asked to check these for us.

Reviewer #1 (Remarks to the Author):

The authors have done an excellent job responding to reviewer 2's queries and concerns. Based on the comments of reviewer 2 I have only one additional suggestion (besides the suggestions in my earlier re-review) for further improvement of the manuscript. Otherwise I think this manuscript is suitable for publication.

Given that on average mule deer arrive on their summering grounds 6 days around the peak of green-up, it appears that timely arrival may be just as important as catching up with the green wave en route.

While this may not be the most novel result and possibly amply discussed in other papers I understand that not too much attention is placed on this result. Yet, I think it would be good to shortly discuss this result given the striking pattern. In the rebuttal to reviewer 2 the authors give some more information on the annual cycle of mule deer, where they state that mule deer give birth on the summering grounds. In this light it would be good to discuss how female mule deer finance the lactation, whether they do so using capital energy stores or locally acquired resources.

I would expect the latter, which would add to the importance of arriving on the summering ground close to the peak of green-up.

Reviewers Comments:

Reviewer #1 (Remarks to the Author):

Comment: I have reassessed the manuscript by Ortega et al. as now submitted to Nature Communications. I find the explanations of the authors to my previous queries to be very satisfactory, and their resulting revisions to have resulted in an improved manuscript. I have only two remaining comments. I consider that these comments are rather minor, and that otherwise this manuscript would now be suitable for publication in Nature Communications. I want to congratulate the authors on this interesting and nice manuscript, which is an excellent example on the flexibility of migration strategies.

Response: Thank you for taking the time to review our manuscript and provide thoughtful and constructive feedback. We are glad that you found our responses and edits satisfactory. We have addressed your two remaining comments by adding an additional paragraph in the discussion and revising Figure 2.

Comment: As the authors rightfully point out, the migration system of birds may be very different from that of ungulates, including differences in how resources are taken up and used during migration. This shows that while similar patterns as found by the authors have been shown in avian species, the findings of the authors are very much novel given that this has not before been shown in ungulates. I think it would be interesting to spend some more discussion on differences in migration systems and strategies. A recent paper by Evans and Bearhop (2022) in the Journal of Animal Ecology pointed out consistent differences between migration strategies with respect to resource use, namely the gradient between 'income migrants' which forage while they travel and 'capital migrants' which store energy reserves prior to departure. Some of the avian migrants which have been shown to be able to compensate for temporal drift belong in the latter category, while mule deer very much belong in the former category. I think it is interesting to discuss the results and potential differences in the ability to compensate for temporal drift between these migration strategies. Maybe this could be discussed in one or several sentences in paragraph 1. 217 - 232.

Response: This is a good point. Thank you for the suggestion! We agree that an income-based versus capital-based strategy could largely influence an animal's behavioral flexibility and ability to compensate. We have added a new paragraph in the discussion (lines 234–243) to discuss the potential importance of behavioral compensation for animals with an income-based strategy. We also clarified that mule deer are both capital and income breeders that rely on fat reserves and available forage to finance reproduction on summer range.

Comment: In my previous review of this manuscript I considered that green-wave surfing en route was not well tested in this manuscript. I thank the authors for their clear explanation that in fact they did test this. On this topic, I have a question or suggestion for figure 2, in which green-wave surfing during migration is shown by displaying Loess regression curves of late, mid- and early migrants. I find that while the Loess curves might work well for visualisation purposes, they do not give a good idea on how the actual data looks like (which might be much more messy indeed), nor are they used to test green-wave surfing as far as I understand from the

manuscript. I would suggest to think of alternative ways for displaying the data. Maybe the lines of individual mule deer tracks could be added to the plot.

Response: Thank you for this suggestion. We agree that the loess regression does obscure some of the details of the tracking en route. To address this comment, we have modified Figure 2 to include the average days from peak instantaneous rate of green-up (IRG) \pm 95% confidence intervals for each category (early, mid, late) and km of the migratory route. We added this information to Figure 2, in addition to the loess regression, because we used the loess regression to determine when early and late migrants began catching up with the green wave during migration.

Reviewer #2 (Remarks to the Author):

****Reviewer #2 was unable to check the responses to their comments so reviewer #1 was asked to check these for us.**

Comment: The authors have done an excellent job responding to reviewer 2's queries and concerns. Based on the comments of reviewer 2 I have only one additional suggestion (besides the suggestions in my earlier re-review) for further improvement of the manuscript. Otherwise I think this manuscript is suitable for publication.

Response: Thank you for taking the time to review our responses to Reviewer 2. We are glad that you found our responses and edits satisfactory. We have addressed your comment for this review and your earlier re-review by adding an additional paragraph in the discussion.

Comment: Given that on average mule deer arrive on their summering grounds 6 days around the peak of green-up, it appears that timely arrival may be just as important as catching up with the green wave en route. While this may not be the most novel result and possibly amply discussed in other papers I understand that not too much attention is placed on this result. Yet, I think it would be good to shortly discuss this result given the striking pattern. In the rebuttal to reviewer 2 the authors give some more information on the annual cycle of mule deer, where they state that mule deer give birth on the summering grounds. In this light it would be good to discuss how female mule deer finance the lactation, whether they do so using capital energy stores or locally acquired resources. I would expect the latter, which would add to the importance of arriving on the summering ground close to the peak of green-up.

Response: This is a good point and to address this comment (along with your previous comment), we have added a new paragraph in the discussion (lines 234–243) to (1) clarify that mule deer are both capital and income breeders that rely on fat reserves and available forage to finance reproduction on summer range, (2) discuss the potential importance of behavioral compensation for animals with an income-based strategy, and (3) point out the importance of arriving on summer range close to peak green-up.